# Tumor Microenvironment of Hepatocellular Carcinoma: Challenges and Opportunities for New Treatment Options

**DOI:** 10.3390/ijms23073778

**Published:** 2022-03-29

**Authors:** Zuzanna Sas, Ewa Cendrowicz, Isabel Weinhäuser, Tomasz P. Rygiel

**Affiliations:** 1Department of Immunology, Medical University of Warsaw, 02-091 Warsaw, Poland; zsas@wum.edu.pl; 2Mossakowski Medical Research Centre, Polish Academy of Sciences, 02-106 Warsaw, Poland; e.cendrowicz@gmail.com; 3Department of Experimental Hematology, Cancer Research Centre Groningen, University Medical Centre Groningen, University of Groningen, 9713 GZ Groningen, The Netherlands; i.weinhauser@umcg.nl; 4Department of Internal Medicine, Medical School of Ribeirao Preto, University of São Paulo, Ribeirao Preto 14015-010, Brazil; 5Centre for Cell Based Therapy, University of São Paulo, Ribeirao Preto 14051-060, Brazil

**Keywords:** hepatocellular carcinoma, tumor microenvironment, cancer therapy, immunotherapies

## Abstract

The prevalence of liver cancer is constantly rising, with increasing incidence and mortality in Europe and the USA in recent decades. Among the different subtypes of liver cancers, hepatocellular carcinoma (HCC) is the most commonly diagnosed liver cancer. Besides advances in diagnosis and promising results of pre-clinical studies, HCC remains a highly lethal disease. In many cases, HCC is an effect of chronic liver inflammation, which leads to the formation of a complex tumor microenvironment (TME) composed of immune and stromal cells. The TME of HCC patients is a challenge for therapies, as it is involved in metastasis and the development of resistance. However, given that the TME is an intricate system of immune and stromal cells interacting with cancer cells, new immune-based therapies are being developed to target the TME of HCC. Therefore, understanding the complexity of the TME in HCC will provide new possibilities to design novel and more effective immunotherapeutics and combinatorial therapies to overcome resistance to treatment. In this review, we describe the role of inflammation during the development and progression of HCC by focusing on TME. We also describe the most recent therapeutic advances for HCC and possible combinatorial treatment options.

## 1. Introduction

Hepatocellular carcinoma (HCC) is the most common form of liver cancer and the second most lethal type of cancer after pancreatic cancer [1]. HCC is responsible for increasing cancer-related mortality [2]. The World Health Organization predicts that by 2025, >1 million people will be diagnosed with liver cancer annually. Current health data indicate a two- to four-fold higher incidence rate in men than in women. The main risk factors include hepatitis B virus (HBV) or hepatitis C virus (HCV) infection, alcoholism, liver fibrosis, metabolic syndrome, and exposure to carcinogens such as aflatoxins from contaminated food crops [3]. The subsequent unresolved persistent inflammation promotes the development of fibrosis, cirrhosis, and HCC [4]. Most HCC cases caused by chronic HBV infection and ingestion of aflatoxin B_1_ occur in Asia and Africa. In contrast, in western countries, where the general HCC incidence rate is lower, HCC is mainly attributed to HCV infection, alcoholism, and non-alcoholic fatty liver disease (NAFLD) [5].

Carcinogenesis-associated inflammation leads to the accumulation of immune cells within the tumor and the surrounding tissues, contributing to tissue remodeling and impairment of their functions [6]. Those immune cells (e.g., T lymphocytes, macrophages, neutrophils, dendritic cells) together with non-immune components, such as fibroblasts, endothelial cells of blood and lymph vessels, and the extracellular matrix (ECM), constitute the immediate surrounding of cancer cells, known as the tumor microenvironment (TME) [7,8,9]. Cells of the TME may have a dual role in tumor development and progression. On the one hand, the TME is responsible for immune surveillance and immunoeditingOn the other hand, the TME may also facilitate invasive tumor growth, colonization of distant organs, or escape from immune surveillance [10,11].

Understanding HCC development and progression is crucial to designing effective HCC-targeting systemic therapies and overcoming resistance to already developed therapies. The type of drug administrated, and its effectiveness, will depend on the cancer stage and the subtype of liver cancer. The main treatment options for HCC include definitive therapies, such as whole or partial hepatectomy and liver transplant. When surgical treatment is not an option, chemical, radiofrequency, microwave ablation, or transarterial chemoembolization are performed [12,13]. Liver transplantation and ablation are potentially curative, but typically only early-stage patients benefit from this type of treatment [14]. Surgery is not always possible due to tumor size and the number of nodules and applies mainly to early diagnosis [15,16]. However, most patients are diagnosed at an advanced stage, due to the lack of tumor-specific symptoms in liver cancers [15]. For these patients, systemic treatments are mostly applied. These include sorafenib and other multi-kinase inhibitors, which are nowadays a standard of care for HCC patients. Conventional chemotherapeutics, which are used in many solid tumors, have been studied for the treatment of HCC with no success as systemic treatment. An example is a doxorubicin, which, in clinical trials, resulted in higher toxicity and no improvement of overall survival (OS) and progression-free survival (PFS) when combined with sorafenib compared to sorafenib alone [17].

Despite many efforts, the therapeutic outcome for HCC patients remains unsatisfactory, and resistance to treatment is frequent. One factor that has been reported to contribute to therapy resistance is the TME. In the current review, we will describe the components of the TME in primary HCC, including the specific role of TAMs, provide an overview of the role of the TME in HCC progression, and layout the recent advances in HCC-targeted therapies.

## 2. Initiation and Development of HCC

HCC is an inflammation-related type of cancer, as more than 90% of HCC cases are a result of hepatic injury and chronic inflammation [18]. Prolonged and unresolved inflammation leads to the infiltration of immune cells into the liver to facilitate tissue remodeling [19]. Numerous studies on the biology of cancer, recently described by Wu et al., confirm that the interaction between cancer cells and their microenvironment influences cancer progression, and results in inhibition of the anti-cancer response by activating and mobilizing immune cells with immunosuppressive properties [20]. As a result of this interaction, tumor cells adapt immune evasion mechanisms and promote the reprogramming of immune cells to support tumor development [21]. The infiltration of immune cells leads to an imbalance of chemokine and cytokine production, and an increased generation of reactive oxygen species (ROS) and reactive nitrogen species promoting the development of fibrosis, cirrhosis, and, eventually, the malignant transformation of liver cells [22]. The formation of such a TME plays a critical role in HCC tumor emergence, progression, and response to therapies.

The immune composition of the TME may differ among HCC patients. Recently, Gao et al. identified four subclasses of HCC based on the TME composition: immune desert (C1), immunogenic (C2), innate immune (C3), and mesenchymal (C4) subclass. The C1 subclass is associated with a better prognosis, and lacks priming T cells, whereas the C2 shows a low pathological stage and is characterized by the infiltration of macrophages, CD4^+^, and CD8^+^ T cells, and B cells. C3 is characterized by the presence of activated immunosuppressive M2 macrophages and displays the worst clinical outcome. In C4, activated CAFs promote epithelial–mesenchymal transition (EMT), which is associated with HCC progression [23,24].

Below, we describe the major cellular components facilitating the emergence and progression of HCC.

## 3. Components of the TME in HCC

### 3.1. Non-Immune Components

*CAFs*. Liver fibrosis is a characteristic hepatic premalignant lesion that underlies around 80–90% of all HCC cases [25]. Under physiological conditions, fibrogenesis is a reversible process initiated by hepatocyte stellate cells (HSC) in response to tissue injury to promote wound healing [26,27]. During this process, ECM, including non-fibrillar collagen type IV and VI, is produced and deposited in the perisinusoidal space (Disse) around the damaged area [26]. Yet, under persistent liver injury, HSCs, which are characterized to be quiescent vitamin-A-rich cells, can acquire a fibrogenic phenotype known as myofibroblastic cells [28]. The emergence of myofibroblasts leads to an increase in ECM production and the accumulation of fibrogenic type I and II collagen, promoting liver fibrosis [29]. Similar to myofibroblasts, cancer-associated fibroblasts (CAFs) derive from HSCs or, in other cases, from tumor cells themselves, driven by epithelial–mesenchymal transition (EMT) [30]. In addition to the secretion of ECM proteins, CAFs secret growth factors, such as epidermal growth factor (EGF) or platelet-derived growth factor (PDGF), immunomodulating chemokines and cytokines, as well as different metalloproteinase (MPP) enzymes (Figure 1) [31]. CAFs can propel tumor-promoting inflammation. Recently, Liu et al. showed an increased expression of the chemotactic cytokine CXCL11 by CAFs, which recruits T cells into inflammatory sites to support the self-renewal of tumor-initiating cells (Figure 1) (Table 1) [32,33]. Furthermore, it was shown that CAFs secrete high amounts of IL-6, which activates the Notch signaling pathway, enhancing stem-cell-like properties of HCC cells (Figure 1) (Table 1) [34]. The stemness of HCC cells is also supported by hepatocyte growth factor (HGF) secreted by CAFs (Figure 1) (Table 1), which binds to its specific receptor c-Met, resulting in cell cycle progression, abnormal proliferation, cell regeneration, and migration [35]. Finally, CAFs also promote tumorigenesis by expressing proangiogenic cytokines, such as vascular endothelial growth factor (VEGF), PDGF, and angiopoietin-1 [36] to facilitate angiogenesis (Figure 1) (Table 1).

*Liver sinusoidal endothelial cells (LSECs)* are specialized fenestrated endothelial cells without a basement membrane lining the hepatic sinusoids. LSECs form a barrier between parenchymal cells and sinusoidal capillaries, crucial for maintaining immune and metabolic homeostasis [37]. Their primary function is the clearance of blood-born waste from systemic circulation and the digestive tract by filtration and endocytosis [38]. During some pathological conditions, LSECs undergo capillarization by losing their fenestration and developing a basement membrane [39]. Under physiological conditions, fenestrated LSECs promote liver stem cell quiescence, whereas capillarized LSECs initiate stem cell activation through the release of PDGF (Table 1), and decreased expression of vasoprotective Kruppel-like factor 2 (KLF2), preceding liver fibrosis, whereas in a normal state, fenestrated LSECs keep liver stem cells in quiescence [40]. The crosstalk between LSECs and other cells of the HCC TME is crucial for the progression of liver fibrosis and, subsequently, HCC development. Viral and bacterial infections affect morphological changes of LSECs, and stimulate the release of proinflammatory cytokines (IL-6, TNF-α) (Table 1), while reducing the production of immunosuppressive cytokines, such as IL-10, TGF-β, and PGE_2_, thereby disturbing liver tolerance (Figure 1) [41]. Limmer et al. showed that LSECs are a non-myeloid cell population able to the cross-present exogenous antigens. In contrast to cross-presentation of antigens by professional antigen-presenting cells, LSECs lead to tolerance, rather than activation, of CD8^+^ T cells [42]. LSECs promote CD8^+^ T cell tolerance development towards antigens from apoptotic tumor cells, contributing to tumor immune escape [43]. It was also observed that during HCC, LSECs overexpress PD-L1 [44] to inhibit T cell activity and support tumor immune evasion [45].

### 3.2. Immune Components of TME in HCC

*T lymphocytes*. Heterogeneous subsets of T cells, including cytotoxic CD8^+^ T lymphocytes (CTLs) and CD4^+^ CD25^+^FoxP3^+^ T regulatory lymphocytes (Tregs), have been identified to infiltrate HCC tumors [46,47]. Whereas CTLs exhibit anti-tumorigenic properties, Tregs promote immunosuppression [48]. CTLs recognize tumor-specific antigens presented by major histocompatibility complex class I (MHC I). Once the target cell has been identified, CTLs release cytotoxic enzymes and cytokines, such as perforins, granzyme B, and IFN-γ, to induce apoptosis [46,49]. Conversely, Tregs counteract anti-tumor response by suppressing CTLs and promoting immune tolerance against neoplastic cells (Figure 1). Studies in HCC indicate that the presence of high numbers of CTLs in combination with a low amount of Tregs predicts improved OS and disease-free survival, whereas CTLs or Tregs alone had no predictive value [50]. Another study published by Fu et al. observed a positive correlation between Tregs’ infiltration and disease progression due to the inhibition of CTLs’ cytotoxic function and proliferation [51]. In relapse HCC samples, T cells present immune dysfunction, as well as low cytotoxicity, contributing to an immunosuppressive microenvironment [52]. Metabolic changes promoted by cancer cells can be partly the reason for T cell dysfunction. Previous studies showed that the release of kynurenine by cancer cells in the TME can lead to T cell exhaustion (Table 1) [53,54,55]. In another study published by Hung et al., the authors delineate the effect of HCC methionine recycling pathways on T cells, where an increase of the metabolites S-adenosyl-L-methionine (SAM) and methylthioadenosine (MTA) remodels chromatin accessibility in T-cells, leading to T cell exhaustion (Figure 1) (Table 1). Treatment of healthy CD8^+^ with SAM or MTA significantly decreased the proliferation rate of T cells and the production of cytotoxic cytokines, and increased T cells exhaustion by upregulating PD-1 and TIM-3. Accordingly, in vivo, the knock-out (KO) of methionine adenosyltransferase 2, which catalyzes the conversion of methionine to S-adenosylmethionine, resulted in a decreased tumor growth of Hep-55.1 cells, and a significant reduction of exhausted T cells [56]. Recently, Bao et al. have shown that high levels of TGF-β enhance the expression of inhibitory receptors, including PD-1 and cytotoxic T cell antigen 4 (CTLA-4) on T lymphocytes in HCC (Table 1) [57]. The PD-1 Ligand (PD-L1) is expressed on different cells of the TME, including tumor cells, Tregs, CAFs, TAMs, and dendritic cells. PD-1/PD-L1 interaction transfers inhibitory signaling to T cells, impairs T cells proliferation, differentiation, and activation by inhibition of MAPK/ERK and PI3K/AKT pathways, which are crucial for cell cycle maintenance [58]. CTLA-4 competes with CD28 for binding to CD80/CD86, disrupting the stimulatory signaling for T cells proliferation [59]. The PD-1 and CTLA-4 immune checkpoint pathways negatively regulate T cells activation to maintain immune tolerance, facilitating the tumor cell evasion from immune cells.

*NK cells* are the key players in immune surveillance against infectious diseases and tumors. They kill pathogen-infected cells by cytolysis, leading to Fas/FasL-mediated apoptotic death [60]. Intra-hepatic NK cells play an important role in HBV and HCV infections underlying HCC. In addition, NK cells prevent fibrosis by the induction of liver stem cell apoptosis. Several studies detected a decreased frequency and impaired function of NK cells facilitating cancer escape from immune surveillance in HCC [61,62,63]. The lower frequency of NK cells is correlated with the expansion of Tregs and increased levels of the immunosuppressive cytokine IL-10 [64]. Early activated liver stem cells express ligands for NK cell activating receptors NKG2D and NKp46, leading to cytotoxic attack of NK cells against liver stem cells [65,66]. The functional impairments of NK cells in HCC, including defective cytokine secretion and production of perforins and granzymes, can be triggered by cells of the TME, including CAFs, myeloid-derived suppressor cells, or TAMs [67]. CAFs produce significantly more immunosuppressive PGE_2_ and IDO enzyme when they are in contact with NK cells, inhibiting their activation and cytotoxic activity (Figure 1) (Table 1) [68]. Moreover, Hoechst et al. reported that myeloid-derived suppressor cells impair the degranulation and secretion of IFN-γ by blocking the NKp30 receptor expressed on the surface of NK cells [69,70]. Monocytes and macrophages derived from TME express the D48 ligand, which interacts with the 2B4 receptor. The 2B4 receptor mediates restricted killing by NK cells, and its interaction with D48 causes NK cell exhaustion and apoptosis (Figure 1) [63]. Moreover, PD-1 is highly expressed on tumor-infiltrating NK cells in HCV and HCC patients. Similar to T cell function, its interaction with PD-L1 blocks the activation of the PI3K/AKT pathway in NK cells, affecting their responsiveness to cytokines and cytotoxic functions [71].

*Macrophages in NASH and Fibrosis.* Non-alcoholic steatohepatitis (NASH) is a progression of NAFLD caused by excess fat deposit, inflammation, ballooning injury, and fibrosis [72]. Macrophages are frequently one of the most abundant immune cells found in the TME [73]. Thanks to their inherent plasticity, macrophages are apt to execute a wide range of functions, which will depend on the signals elicited by their environment [74]. Within the liver, the largest macrophage population consists of Kupffer cells (KCs), whereas monocyte-derived macrophages (MoMΦs) can infiltrate the liver in the event of inflammation [75]. In contrast to MoMΦs, which are recruited from the bone marrow [75], KCs originate from the yolk-sac-derived CSF1R^+^ erythromyeloid progenitors [76], and possess self-renewal capacities [77]. KCs undergo classical and non-classical polarization to M1 and M2 macrophage subtypes in response to chronic liver injury. The balance of the M1/M2 KC-derived macrophages is important in the regulation of chronic liver inflammation [78].

Depletion studies of macrophages in mice fed with a high-fat diet showed a decrease in steatosis, suggesting that macrophages contributed to the unfolding of NASH [79,80]. Daemen et al. describe a loss of KCs characterized by the expression of TIM4, and an increase of TIM4^neg^ monocyte-derived macrophages, suggesting the infiltration of monocytes during the emergence of NASH [81]. Another study, published by Krenkel et al., evaluated the role of infiltrating MoMΦs during the development of NASH, and the effect of cenicriviroc (CVC), a CCR2/CCR5 antagonist. Biopsy analyses confirmed a positive correlation between CCR2^+^ macrophage infiltration and NASH/fibrosis progression. Furthermore, mice on a high-fat diet treated with CVC presented a decreased incidence of steatohepatitis, fatty degeneration, and frequency of infiltrating macrophages compared to control mice. Additional transcriptome analysis of MoMΦs and KCs in the context of NASH indicated an increased expression of inflammatory cytokines, such as IL-1β and TNF-α, and ECM proteins, contributing to liver fibrosis [82]. The pro-inflammatory phenotype of macrophages in NASH context was also confirmed by single-cell RNA sequencing (scRNA-seq) of murine diet-induced NASH livers compared to control [83]. Furthermore, scRNA-seq analysis identified two KC populations characterized by Trem2^high^ and Trem2^low^ expression, where the former was specific for NASH livers. Conversely, the expression of Trem2 and other NASH-associated genes, such as Gpnmb, Tnf, and Col1a1, were downregulated when mice were treated with Elafibranor. In accordance with the murine data, Trem2 was also upregulated in hepatic steatosis and NASH patients and correlated with disease severity.

Mechanisms that underlie the progression of NASH and fibrosis by macrophages are still not fully understood. Amongst others, the depletion of macrophages in a high-fat diet-induced murine NASH model results in the decrease of inflammatory cytokines, such as IL-1β, IL-15, complement component C1, and Ccl6, whereas the fatty oxidation gene peroxisome proliferator-activated receptor alpha (PPARα) was increased, resulting in a decrease of liver triglyceride and glucosylceramide. The study further confirmed that the administration of IL-1β in vivo decreased the expression of PPARα in hepatocytes, explaining, in part, the effect of macrophages on steatosis [84]. Furthermore, other macrophage depletion studies confirmed a decrease in inflammation, along with a diminished expression of fibrosis-related genes and oxidative damage [85,86,87,88]. Finally, it has been suggested that MerTk protein expressed by macrophages, and activated by its ligand Gas6, leads to an increase in TGF-β synthesis and release, activating HSCs and promoting fibrosis in NASH murine models. Conversely, cleavage of MerTk reduced NASH fibrosis. Furthermore, the authors observed enhanced enzymatic activity of ADAM17, which cleaves MerTk, in mice receiving a NASH diet for 8 weeks, corresponding to a steatosis state, compared to mice being fed for 16 weeks with a NASH diet. The authors suggest that this is due to a decrease of all-trans-retinoic acid (ATRA) during NASH progression, which activates P38, and which, in turn, activates ADAM17 [89].

*Macrophages in HCC.* Previous reports which aimed to identify the impact of macrophages on HCC development identified CD68^+^ macrophages to confer poor prognosis [90], whereas CD68^+^CD169^+^ macrophages were associated with improved OS [91]. Recent studies employed scRNA-seq analysis to unravel the heterogeneity of hepatic macrophages. Analysis of the healthy liver by MacParland et al. suggests the presence of two macrophage subpopulations, CD68^+^ MARCO^+^ and CD68^+^ MARCO^−^ macrophages, characterized by an anti- and pro-inflammatory phenotype, respectively [92]. Yet, scRNA-seq analysis on blood-derived and liver-derived macrophages indicated that the latter portrayed an anti-inflammatory profile when compared to blood-derived macrophages [93], reinforcing ontogenetic differences. Furthermore, Sierro et al. characterized MHCII^+^ expressing cells, which were previously believed to be dendritic cells, as a distinct MoMΦs entity located in the hepatic capsule, essential to fight bacterial infections [94]. Overall, it becomes clear that the liver is home to a diversity of macrophages executing a variety of functions, which could be exploited by the emergence of HCC. Within this context, Zhang et al. identified a macrophage population enriched for TAM signatures, which predicted poor prognosis in HCC patients. Upregulated genes associated with the TAM signature included the ferroportin transporter SCL4OA1, and the type I transmembrane glycoprotein GPNMB, which were mutually exclusively expressed by different macrophage cells [95]. In vitro studies generating a knock-out of both genes in THP1-derived macrophages resulted in the downregulation of inflammatory cytokines, such as TNFα, IL-23, IL-6, and IL-12p40, whereas the production of IL-1β was increased [95]. Another study conducted by Sun et al. aimed to understand the transcriptional differences of immune cells in primary HCC tumors compared to early relapse samples. The authors identified eleven myeloid clusters, of which five were macrophage clusters distinct from resident KCs. Additionally, relapse HCC samples presented an increase of gene signature enriched for immune evasion mechanisms, including the upregulation of *CD47* and *HLA* genes [52]. It is worth mentioning that both studies detected an overlap in M1 and M2 signatures, suggesting a much higher complexity of macrophage biology, which goes well beyond the simplistic view of M1 and M2 macrophages [52,95].

Functionally, TAMs promote tumor progression via several mechanisms by promoting cancer cell proliferation, angiogenesis, dissemination, and immune evasion [96]. The inflammatory environment of HCC leads to an upregulation of cytokines and chemokines, including CCL2. CCL2 recruits monocytes via the CCL2 receptor (CCR2) (Table 1). Once monocytes reach the liver tissue, they can differentiate into macrophages (Figure 1) [97]. In vivo, CCR2^+^ macrophages have been detected to localize close to endothelial cells at the border of the HCC tumor to initiate angiogenesis [98,99]. Moreover, the blockage of CCR2 inhibited macrophage liver infiltration, which, in turn, resulted in a reduction of tumor growth [97,98] due to an increase of cytotoxic CD8^+^ present at the tumor site [97]. In a study published by Zhang et al., the authors demonstrate increased PD-L1 expression detected in HCC macrophages inducing T-cell exhaustion. The expression of PD-L1 negatively correlated with the protein tyrosine phosphatase receptor type O (PTPRO), which was suggested to downregulate PD-L1 expression by activating the Jak2/Stat1 and Jak2/Stat3/c-Myc signaling pathway. Conversely, IL-6 upregulated PD-L1 expression by activating the Stat3/c-Myc signal to increase the transcript levels of miR-25-3p to inhibit PTPRO [100,101]. Though the presence of macrophages can aggravate tumorigenesis, this interaction is not a one-way street, meaning that it is important to understand how macrophages affect tumor cells and vice versa to develop new therapeutic strategies. In a study published by Wang et al., the authors identified a highly proliferative Ki67^high^ macrophage population, which predicted poor OS. The emergence and maintenance of Ki67^high^ macrophages were stimulated by the release of adenosine by tumor cells, whereas the secretion of GM-CSF by macrophages induced the expression of the adenosine receptor A2A. Subsequent binding of adenosine to A2A activated the PI3K/Akt pathway, facilitating the expansion of macrophages [102].

*Neutrophils*. The role of tumor-associated neutrophils (TANs) is often overlooked in discussions about the development and progression of cancer. However, TANs play a significant role in the TME of HCC. Similar to M1 and M2 macrophages, TANs also exhibit two distinct anti-tumorigenic (N1) and protumorigenic (N2) phenotypes [103]. Inflammatory TME in HCC promotes the formation of N2 neutrophils. These include the activity of CAFs, and the presence of high concentrations of TGF-β, GM-CSF, and TNF. N2 neutrophils are characterized by enhanced expression of PD-L1, which, in turn, plays a role in the suppression of T cells in HCC [104]. Various studies have proven that either high infiltration of neutrophils or neutrophil/lymphocyte ratio in HCC can correlate with poor outcomes for the patient (summarized in [103]). The presence of TANs supports the development of HCC via direct crosstalk with cancer cells. They release neutrophil extracellular traps (NETs), which support tumor growth by secretion of MMP-9 and cathepsin G, as well as the induction of metastasis of HCC cells [105]. Therefore, NETs can be potential targets for the new treatment options in HCC [106].

### 3.3. The Role of Cytokines in the Development and Progression of HCC

As a result of the persistent inflammation detected in HCC, several cytokines are upregulated in HCC patients. IL-6 is a pleiotropic cytokine exerting both pro and anti-inflammatory functions, frequently upregulated in different cancers, such as HCC, leukemia, or lung cancer [107,108,109]. IL-6 can either bind to the IL-6 receptor expressed on the cell surface or form a complex with the soluble IL-6 receptor, whichtransmits a signal to gp130 known as “IL-6 trans-signaling”. The function of IL-6 signaling in HCC appears to be two-fold depending on the type of cell releasing IL-6 and the disease stage. In the context of chronic liver inflammation, the loss of IL-6 or STAT3 signaling accelerated HCC development in Mdr2^−/−^ mice. Depleting IL-6 and STAT3 signaling resulted in an accumulation of hepatic steatosis, macrophage recruitment, and increase of hepatocyte proliferation [110]. However, Bergman et al. showed that macrophages release the IL-6 receptor to induce IL-6 trans-signaling, promoting HCC progression in a murine diethylnitrosamine (DEN) model. The authors suggest that IL-6 trans-signaling inhibits DEN-induced cell death mediated by p53 and activates the β-catenin signaling pathway [111]. Targeted deletion of IL-6 in macrophages reduced the incidence of spontaneous liver cancer by disrupting the IL-6/STAT3 axis, promoting cell proliferation and cell death resistance [112].

Another cytokine that affects HCC progression is TGF-β [113,114]. Like IL-6, TGF-β exerts multiple functions by binding to a complex of membrane-bound type I and type II receptors [115]. In HCC, TGF-β represents a double-edged sword, promoting anti-tumorigenic activity in the initial phase of HCC emergence, but adapting pro-tumorigenic functions during HCC progression [116]. A study published by Im et al. showed that heterozygote deletion of the TβR-II gene significantly increased the frequency of neoplastic lesions when mice were treated with DEN compared to control mice [117]. Within this notion, the overexpression of the TGF-β downstream target, Smad3, suppresses DEN-induced tumor growth by downregulating the anti-apoptotic protein, Bcl-2 [118]. Furthermore, TGF-β induces senescence in well-differentiated HCC cells by promoting the accumulation of Nox4 and reactive oxygen species (ROS), leading to the upregulation of p21^Cip1^ and p15^Ink4b^, resulting in HCC G1 arrest in vitro and decreased tumor growth in vivo [119]. Another report, which focuses on liver cancer metastasis, reveals a positive feedback loop between Kruppel-like-factor 17 (KLF17) and TGF-β/Smad3 to prevent HCC progression and metastasis. Additionally, the authors suggest the formation of a transcriptional complex of KLF17 and TGF-β/Smad3 to bind to KL17 response elements and Smad-binding elements in the presence of TGF-β, thereby regulating genes associated with cancer metastasis [120].

However, though many studies suggest TGF-β mediated anti-tumorigenic functions, it has been shown that HCC cells develop mechanisms that allow them to become resistant to the TGF-β mediated tumor-suppressive effects. Within this context, the upregulation of the epidermal growth factor receptor signaling pathway can inhibit the pro-apoptotic function of TGF-β [121]. Furthermore, high expression of TGF-β has been associated with EMT facilitating HCC migration and metastasis (Table 1) [122]. Wang et al. showed that EMT-associated proteins, such as vimentin and Snail, as well as p-Smad2/3 and p-STAT3/STAT3, are upregulated in the HepG2 cells in the presence of TGF-β, which was abrogated upon inhibition of STAT3. Moreover, the inhibition of STAT3 significantly decreased HepG2 cell migration when treated with TGF-β, suggesting TGF-β mediated EMT to be STAT3-dependent [123]. It becomes evident that the dichotomous role of TGF-β has a significant impact on the emergence and progression of HCC.

**Table 1 ijms-23-03778-t001:** TME components and their major effects on HCC development.

Component	Secreted Factors	Effect	Publication
CAFs	CXCL11	recruitment of T helper cells into inflammatory sites to support self-renewal of tumor-initiating cells	[32,33]
	IL-6, HGF	enhancing of HCC cell stemness by activation of Notch signaling (IL-6) and interaction with c-Met receptor (HGF), supporting cell cycle progression and cell regeneration	[34,35]
	VEGF, PDGF, angiopoietin-1	induction of angiogenesis, supporting tumor growth	[36]
	IDO, PGE_2_	suppression of NK cell activation and cytotoxicity, creating favorable environment for tumor progression	[68]
LSECs	PDGF	activation of liver stem cells which can give rise to tumor cells after malignant transformation	[40]
	IL-6, TNF-α	proinflammatory activity	[41]
Cancer cells	Kynurenine, SAM, MTA	decrease of T cells proliferation rate and production of cytotoxic cytokines, leading to T cells exhaustion and failure in cancer elimination	[54,55,56,57]
Macrophages	CCL2	recruitment of monocytes to the tissue where they can differentiate into macrophages	[97]
Malignant hepatocytes	TGF-β	upregulation of the expression of inhibitory receptors PD-1 and CTLA-4, which negatively regulate T cell activation; support of EMT, which facilitates HCC migration and metastasis	[57,122]

Overall, the progression to HCC is mediated by a complex and interconnected interplay of different cell types found in the TME. This interaction leads to a deregulated release of immunomodulating and pro-angiogenic factors; increased ECM production; and secretion of growth factors by CAFs, LSECs, and myeloid cells, facilitating HCC growth, stem cell activation, and dissemination. Furthermore, the remodeling of HCC TME is characterized by a dysfunctional adaptive immune system, whereby the function of CTLs and NK cells is inhibited by monocytes or TAMs, myeloid-derived suppressor cells, and TANs. This interaction leads to the exhaustion of T and NK cells, upregulation of PD-L1 expression by HCC cells, increased frequency of Tregs, and impaired degranulation of NK cells. Hence, it can be concluded that the emergence of HCC entails several alterations in HCC TME that could be therapeutically exploited to improve the clinical outcome of HCC patients.

## 4. Treatment of HCC

Besides many efforts to find an effective therapy for HCC, the prognosis of HCC patients remains dismal. The main reason for the high mortality rate is a late diagnosis, which limits treatment options, and a high recurrence rate after treatment [124]. Therefore, finding effective therapy to suppress the recurrence of HCC is needed. Given that TME plays a role in the development of HCC and resistance to standard therapies, immunotherapies are a promising approach for supportive treatment which can decrease the recurrence of HCC. In vitro cultures of cancer cells are used for drug development. However, this tool is far from perfect in mimicking human TME. To better model TME, three-dimensional (3D) cultures of cancer cells, together with fibroblasts and immune cells, are being developed. These culture systems could be used for preclinical drug discovery. This issue was summarized in the recent review by Nii et al. [125]. Below, we describe standard treatments for HCC, novel immunotherapies, and potential immunotherapeutic options.

### 4.1. Interventional Treatment

One of the most effective therapies is liver transplantation, which cures not only HCC but also underlying liver cirrhosis. It is applied to patients with early unresectable HCC and results in a 70% 5-year survival rate. However, liver transplantations are mainly limited by the number of donors. Therefore, many patients die while waiting for transplantation or become unsuitable for this type of treatment. Surgical resection is possible only for patients with a single nodule and good liver function. Patients with underlying cirrhosis have an increased risk of hepatic decompensation after surgery [126]. Another invasive treatment is transarterial (chemo) embolization. It is a neoadjuvant therapy and the first-line treatment for patients with an intermediate stage of HCC. The method involves occlusion of the tumor arterial blood supply with embolizing agents or with included chemotherapeutic agents. It aims to deliver chemotherapeutic drugs into the tumor arterial supply and, at the same time, to cut off the blood supply of the tumor, which results in starvation and cancer cell death [127]. Recently, drug-eluting beads were developed to decrease systemic toxicity and improve the local release of chemotherapeutic drugs. This method reduced toxicity with a similar effect as standard transarterial chemoembolization [128]. Radiofrequency ablation (RFA) and microwave ablation (MWA) are local treatment methods that produce heat-based thermal cytotoxicity. The main difference between RFA and MWA is the heat source. In RFA, electrical current in the radiofrequency range is applied, whereas, in MWA, the heat is derived from electromagnetic energy [129]. Both methods are used to treat small local areas near the electrode up to 2–3 cm (for RFA) and 5–8 cm (for MWA) in diameter and in patients who are not candidates for surgery. Currently, modifications to the methods are applied, including multipolar electrodes, which enlarge the treatment areas. RFA is most effective in patients with tumors up to 2 cm in diameter and results in a 5-year survival rate of 40–70% [126].

### 4.2. Molecular-Targeted Therapies

#### 4.2.1. Approved Therapies

Sorafenib is a multiple kinase inhibitor which targets VEGFR-1, VEGFR-2, VEGFR-3, platelet-derived growth factor receptor (PDGFR) β, RET, c-KIT, and FMS-like tyrosine kinase-3. Sorafenib was associated with improved OS in the SHARP trial, prolonging OS from 7.9 for placebo to 10.7 months. Since 2007, sorafenib has been considered the standard of care for the first-line treatment of advanced unresectable HCC (Table 2). The downsides of treatments with sorafenib are associated with adverse events, price, and resistance to treatment by almost 50% of patients [130,131]. This resistance is likely caused by some tumor-associated immune cells, such as TANs [132] and TAMs [133]. Therefore, combinatorial therapy with immunotherapy can effectively overcome resistance to sorafenib. Sorafenib was the only systemic treatment option for almost a decade until the results of a randomized, double-blind, placebo-controlled multinational RESORCE trial, which resulted in the approval in 2017 of another kinase inhibitor, regorafenib [134]. It was the first drug to be approved as a second-line treatment for patients who progressed after sorafenib treatment (Table 2). Regorafenib prolonged OS up to 10.7 mo in the treatment group compared to 7.9 mo in the placebo group [130], resulting in similar effectiveness as sorafenib. The most common adverse effects included hand–foot skin reaction, diarrhea, fatigue, anorexia, hypertension, and oral mucositis. Treatment-related adverse effects were 2-fold higher in the regorafenib group than in the placebo group [130]. Regorafenib inhibits a broader spectrum of kinases than sorafenib, including VEGFR-1, -2, -3, TIE2, PDGFR-β, FGFR1, KIT< RET, c-RAF, and BRAF [130], and exhibited stronger anti-tumor activity in mouse models [130]. One year after the approval of regorafenib, lenvatinib, an inhibitor of multiple receptor tyrosine kinases, was approved as a result of the REFLECT trial [135]. It is also the first-line treatment of unresectable HCC and the only systemic-targeting multiple-kinases alternative for sorafenib (Table 2). Lenvatinib improved secondary endpoints compared to sorafenib, such as objective response rates, prolonged progression-free survival, and longer time to progression [135]. Cabozantinib, an inhibitor of tyrosine kinases VEGFR1, 2, 3, MET, and AXL, declares the list of approvals of kinase inhibitors closed. The CELESTIAL clinical trial showed a longer OS of patients compared to placebo (10.2 vs. 8 months), and progression-free survival of 5.2 months compared to 1.9 in the placebo group, and cabozantinib was approved by FDA in 2019 as a second-line treatment for patients treated previously with sorafenib (Table 2) [136]. Other kinase inhibitors, including sunitinib, linifanib, brivanib, tivantinib, and everolimus, did not show survival benefits over sorafenib [137]. The main reasons for treatment failure included liver toxicity and lack of anti-tumor potency, which resulted in the lack of survival benefit over sorafenib [138].

Recently, immunotherapies joined the repertory of molecular-targeted therapies for HCC, such as the combination of monoclonal antibodies atezolizumab and bevacizumab, which target PD-L1 and VEGF, respectively. These antibodies turned out to be successful in the treatment of HCC. The direct and indirect role of VEGF in the development and progression of many types of cancer, including HCC, is well described in the literature [139]. Moreover, the rationale for using PD-1/PD-L1 inhibitors has been known for years and has been approved in many solid tumors and hematological malignancies, including non-small-cell lung carcinoma, melanoma, urothelial cancer, esophageal cancer, renal cell carcinoma, and Hodgkin’s lymphoma [140]. VEGF promotes tumor growth, the proliferation of Tregs, a polarization of macrophages into a more pro-tumorigenic M2 subtype, a release of immunosuppressive cytokines, and inhibition of dendritic cell maturation and antigen presentation [139]. There is strong reasoning to combine PD-1/PD-L1 and VEGF inhibitors. VEGF inhibitors, among others, promote T cell activation in the priming phase, migration of T cells to the tumor site, and transit immunosuppressive TME into more immunostimulatory TME. PD-1/PD-L1 inhibitors, in turn, act during the second phase of T cell activation and further enhance anti-tumor T cell cytotoxicity [139]. Atezolizumab and bevacizumab have been approved as a first-line treatment for unresectable locally advanced or metastatic hepatocellular carcinoma [141] as a result of the IMbrave150 study. In this study, the combination of atezolizumab + bevacizumab vs. sorafenib was investigated, with a very promising outcome (Table 2). Median OS was 19.2 months in the group treated with atezolizumab + bevacizumab vs. 13.4 months with sorafenib. Of note, both PD-1/PD-L1 and VEGF inhibitors had been tested as single agents in clinical trials for the treatment of HCC. Besides promising response rates in phase 1/2, neither of the drugs alone showed sufficient improvement in OS of HCC patients [142,143]. It proves that in the complex microenvironment of HCC, combinatorial therapies are more likely to succeed.

**Table 2 ijms-23-03778-t002:** Approved therapeutics for the treatment of HCC.

Name	Molecular Targets	Treatment Recommendations *	Approval	Clinical Trial Number	Publication
Sorafenib	Multikinase inhibitor that targets VEGFR-1, 2, 3, PDGFRβ, RET, c-KIT, FMS tyrosine kinase-3	1st line in advanced unresectable HCC	2007	NCT00492752	[144,145]
Lenvatinib	Multi-kinase inhibitor targeting VEGFR1, 2, 3, PDGFRα, KIT, and RET kinases	1st line In unresectable HCC	2018	NCT01761266	[146]
Regorafenib	Multi-kinase inhibitor targeting VEGFR-1, 2, 3, TIE2, PDGFR-β, FGFR1, KIT, RET, c-RAF, BRAF	2nd line for patients which progressed after sorafenib treatment	2017	NCT01774344	[134]
Cabozantinib	Multi-kinase inhibitor targeting VEGFR1, 2, 3, MET, AXL	2nd line for patients treated previously with sorafenib	2019	NCT01908426	[136]
Ramucirumab	Monoclonal antibody targeting VEGFR2	2nd line after sorafenib treatment in patients with alpha fetoprotein of ≥400 ng/mL	2019	NCT02435433	[147]
Atezolizumab + bevacizumab	Combination of monoclonal antibodies targeting PD-L1 (atezolizumab) and VEGF (bevacizumab)	1st line unresectable locally advanced or metastatic HCC	2020	NCT03434379	[141,148]

* approval year and treatment recommendations are FDA-based.

#### 4.2.2. Therapeutics at the Development Stage

Another promising immunotherapeutic strategy is the combination of durvalumab (PD-L1 inhibitor) and tremelimumab (CTLA-4 inhibitor). Treatment with these two antibodies recently showed promising results in Phase III HIMALAYA clinical trials (NCT03298451), a multicenter study conducted across 16 countries. Both PD-L1 and CTLA-4 are inhibitory molecules expressed in T cells. Their effectiveness in improving the OS of HCC patients highlights the role of T cells in the treatment of HCC. Durvalumab is also currently investigated as a monotherapeutic agent in a Phase II trial for advanced HCC with active chronic hepatitis B virus infection (NCT04294498).

*γδ T cells*. Another novel option of immunotherapy of HCC is the adoptive T cell transfer of gamma-delta T cells (γδ T cells). Low infiltration of γδ T cells in the peritumoral liver tissue is related to a higher recurrence rate in HCC, and can predict postoperative recurrence [61]. A Phase I clinical trial is currently registered in China for the investigation of γδ T cells for the treatment of advanced hepatitis B-related HCC (NCT04032392). Adoptive transfer of allogenic γδ T cells was also used in combination with local interventional treatment, with distant progression-free survival of eight months for combinatorial therapy vs. four months for locoregional treatment alone [149], with manageable adverse events.

*CAR-T cell therapies* are one of the youngest but most promising discoveries of immunotherapies. Since the engineering of the first-generation CAR-T cells in 1993, many innovations have been introduced into CAR-T cell products. Up to date, five CAR-T cell therapies have been approved, Abecma in 2017 as an orphan drug (for treatment of multiple myeloma), Breyanzi in 2021 (for the treatment of large B cell lymphoma and follicular lymphoma grade 3B), Kymriah in 2014 (for treatment of B cell ALL) and 2016 for treatment of DLBCL, Tecartus in 2019 as orphan medicine for mantle cell lymphoma, and Yescarta in 2014 for large B cell lymphoma. All of the approved CAR-T cell therapies target hematological malignancies. Until now, no successful CAR-T cell has been implemented for the treatment of solid tumors. The main limitations of using CAR-T cells in solid tumors are tumor antigen heterogeneity, trafficking, infiltration into the tumor tissue and immunosuppressive TME [150]. Besides obvious limitations, several CAR-T therapies are currently in clinical trials for HCC, targeting various surface and intracellular antigens, including Glypican-3 (GPC3) [151], c-Met membrane glycoproteins [152], like Mucin 1 (MUC1) [153] and Epithelial cell adhesion molecule (EpCAM) (NCT03013712, NCT02729493), CD133 [154], CD147 [155], and alpha-fetoprotein (AFP) (NCT03349255). These proteins are often overexpressed in HCC, and involved in stem-cell-like characteristics, including self-renewal and differentiation (e.g., EpCAM, CD133), tumor proliferation, and metastasis (e.g., CD133 and CD147, c-Met), or invasion (e.g., GPC3, c-Met). One of the interesting HCC targets for CAR-T cells is alpha-fetoprotein (AFP), which is expressed in HCC patients, but not in the liver of healthy adults. AFP is a soluble protein; therefore, it cannot be targeted by conventional CAR-T cells. However, in a recent study, CAR-T cells selectively recognized AFP158-166 peptides presented on HLA-A tumor cells, which led to the selective elimination of HLA-A/AFP^+^ cancer cells [156]. A recent review by Guo J and Tang Q provides an excellent update on the CAR-T cell therapies in HCC [152]. Another interesting approach that is an alternative to conventional T cell therapies is CAR-T cell targeting GPC-3 engineered from unconventional γδ T cells. These T cells were engineered to express and secrete IL-15. It was shown that GPC-3. CAR/sIL-15 Vδ1 T cells were able to proliferate, accumulate in the tumor [157], and control tumor growth in vitro and in the HepG2 mouse model.

*Dendritic cell vaccines.* DCs are the most powerful type of antigen-presenting cells. They form a bridge between the innate and adaptive immune systems via cross-presentation of antigen to CD8^+^ T cells. Cross-presentation is considered the main step toward boosting the anti-tumor cytotoxic effect of T cells [158]. DCs, as a tool in anticancer therapies, have been studied for many years. The most common practice of preparation of DC vaccines is to isolate CD14^+^ monocytes from patients and culture these monocytes with stimulatory cytokines, such as GM-CSF and IL-4. Next, dendritic cells are loaded with tumor antigens, such as tumor lysates or tumor-associated antigens, and injected into the patient [124]. The first DC vaccine was approved in 2010 for the treatment of advanced prostate cancer [159]. Since then, the same approach was used in clinical trials against many other cancers, but with limited success [160]. Several clinical trials have also been initiated with DC vaccines against HCC. A meta-analysis of 22 clinical trials showed that DC vaccines improve OS after one year, and the combination of DC vaccines with cytokine-induced killer cells (CIKCs) improved OS after one and five years [158]. Besides improvement in OS, DC vaccines also prolonged recurrence time, and reduced tumor size, with a reasonable safety profile. Recent pre-clinical studies have demonstrated further improvement of HCC treatment when using a combination of tumor-specific DC vaccines with PD-1 blocking drugs [161,162]. The main reason for using this combinatorial approach is the integration of T cell activation by cross-presentation of antigens with unlocking PD-1/PD-L1 interaction, which inhibits T cells, resulting in overall enhanced anti-tumor immunity. In the pre-clinical mouse model, a single therapeutic approach (either DC vaccine or PD-L1 blockade) resulted in better OS, lower tumor volume, and higher tumor apoptosis compared to placebo. The combinatorial approach resulted in further enhanced effects compared to single treatment effects [161].

*Targeting complement system*. C5a is a member of the complement system of plasma proteins which plays an important role during an innate immune response. Complement is also engaged in the development of various diseases, including cancer. C5a is a chemoattractant that plays an essential role in the TME of lung cancer patients. C5aR is also highly expressed in HCC cell lines and tissues, and upon binding to C5a, it promotes HCC cell invasion, migration, and expression of EMT-related markers [163]. C5a formation is indirectly promoted by complement factor H (CFH). CFH regulates the activation of the alternative pathway of complement activation, which, in turn, is involved in the activation of the C5 convertase. C5 convertase cleaves C5 into C5a and C5b [164]. A recent study showed that CFH-deficient mice spontaneously developed hepatic tumors [165]. Moreover, the increase in CFH expression is associated with better OS in patients with HCC, and mutations in the CFH gene are associated with decreased survival. High levels of C5a were also detected in the plasma of patients with advanced (stage III) tumor stage compared to early tumor stage, and high levels of C5a were correlated with large tumor size [166]. Increased expression of C5a is also reversely associated with the expression of TGFβR3 in HCC, and a low expression level of TGFβR3 is observed in HCC tissues [166]. Therefore, targeting C5a/C5aR interaction or upstream pathways regulating C5a expression may be a promising approach to target HCC cells.

### 4.3. Resistance to Therapies and Combinatorial Approaches to Overcome Resistance

Intrinsic and acquired resistance to therapies is one of the main reasons for the poor outcomes of HCC patients. Many different processes and components of the TME play a role in the development of resistance and are still under investigation. One of the mechanisms of resistance involves hypoxia-inducible factors (HIFs). In the first stages, sorafenib inhibits HIF-alpha production, which, in turn, results in the inhibition of VEGF and lower vascularization. This positive effect is short-term, and prolonged treatment with sorafenib leads to the increase of tumor hypoxia and the selection of cell clones that are more resistant to the deficit of oxygen and nutrients [167]. Combinatorial therapies of sorafenib and HIF-targeting inhibitors, such as ICI-118,551, melatonin, or genistein, are potential synthetic and natural compounds which can help overcome developed resistance [167]. Immune components known for tumor-promoting roles are TAMs [78]. The immunosuppressive function of TAMs can affect treatment by other drugs, which should directly or indirectly activate anti-cancer immune defense. In fact, M2-type macrophages cause tumor resistance to sorafenib by continuous promotion of tumor growth and metastasis, and secretion of HGF [133]. Therefore, targeting TAMs in HCC is a promising approach for combinatorial therapies with currently known therapeutic and immunotherapy, as well as novel drugs which target both immune and cancer cells in HCC. Two main approaches to target TAMs include reduction of the number of TAMs in the TME or re-education of TAMs into a more pro-inflammatory M1 subtype. The first approach can be achieved by either inhibition of recruitment of monocytes to the TME or depletion of infiltrated TAMs. Both methods were described in more detail in our recent review [168]. One potential target for reducing macrophage infiltration in HCC is the CCL2/CCR2 pathway. CCL2 is overexpressed and is a prognostic factor in hepatocellular carcinoma patients [78]. It is a chemokine that attracts CCR2-expressing monocytes to the tumor side. It was shown that CCL2-targeting antibodies reduced tumor growth and inhibited angiogenesis or metastasis in pre-clinical studies of several cancer types. However, clinical studies of several CCL2- and CCR2-neutralizing antibodies, such as carlumab, plozalizumab, and CCX872-B, did not show efficacy in clinical trials in various cancer types. However, CCL2/CCR2-targeting antibodies can be promising therapeutics for combinatorial therapies in HCC, where macrophages play an immunoregulatory function. The addition of CCR2 natural inhibitor 747 in the pre-clinical animal model potentiated the therapeutic activity of sorafenib by increasing the number of CD8^+^ T cells as a consequence of the depletion of immunosuppressive macrophages [169,170]. Other chemokines that attract monocytes to the tumor are also being targeted by drugs that aim to reduce macrophage number in tumors in some malignancies. CCR5/CCL5 may be another potential target chemokine in HCC [169]. CCR5 is a monocyte and lymphocyte receptor involved in tumor growth, resistance to therapies, and polarization of macrophages to immunosuppressive subtypes. CCR5 and CCL5 are overexpressed in HCC tissues compared to non-neoplastic liver tissues [169]. Drugs that target the CCL5/CCR pathway include monoclonal antibodies, such as leronlimab, and small molecule inhibitors, such as maraviroc, vicriviroc, etc. [168]. Similar to TAMs, TANs also cause resistance to sorafenib in HCC. A high TAN/lymphocyte ratio often correlates with relapse after treatment and shortens progression-free survival and OS after interventional therapy. It was also shown that TANs are involved in the development of resistance to sorafenib treatment by recruiting macrophages and Tregs to the TME of HCC, which leads to cancer progression [132]. Given the complex TME in HCC, combinatorial therapies seem the most appropriate approach to improving current therapies. Targeting TAMs or TANs together with sorafenib treatment may improve the effectiveness of this drug. On the other hand, combiningconventional therapies with an anti-inflammatory diet or food-derived phytochemicals has also gained attention. Flavonoids, such as curcumin, luteolin, resveratrol, or quercetin, may play a chemopreventive role in HCC. Resveratrol had an anti-proliferative and pro-apoptotic effect on HCC cells [171] and suppressed the cross-talk between cancer cells and TME [172].

## 5. Discussion

Overall, it becomes evident that both intrinsic and extrinsic factors contribute to the dismal outcome of HCC patients. Within this review, we described the intricate interplay between the TME and HCC cells by delineating the disease-promoting function of several cellular entities within the HCC TME. The subsequent result of this interaction is the formation of a tolerogenic TME, allowing a more aggressive course of the disease. Though new therapeutic approaches, such as the administration of atezolizumab together with bevacizumab, are associated with improved OS compared to sorafenib, these differences remain limited. Given the crucial role of the TME during HCC emergence, progression, and recurrence, several new therapeutic strategies in the form of immunotherapy are being investigated in clinical trials, such as tremelimumab (CTLA-4 inhibitor), the transfer of γδ T cells, or CAR-T cell therapies. Single-agent therapies did not result in satisfactory outcomes, possibly due to the complexity of the TME in HCC. Therefore, it can be concluded that combinatorial therapies will be the approach of choice for HCC. Combinatorial therapies can include targeting immunosuppressive cells in the TME, such as macrophages and neutrophils, or HIFs. Currently, many drugs are being developed, including synthetic drugs, biological therapeutics, vaccines, and natural compounds. The influence of diet, and the role of phytopharmaceuticals in HCC, has also been highlighted [173]. Therefore, a more holistic approach during therapy should be considered a combinatorial option with current therapies. Prospectively, more therapeutic agents will be approved for HCC, and personalized treatment schemes will be needed to achieve the maximum clinical benefit for the patients.

## Figures and Tables

**Figure 1 ijms-23-03778-f001:**
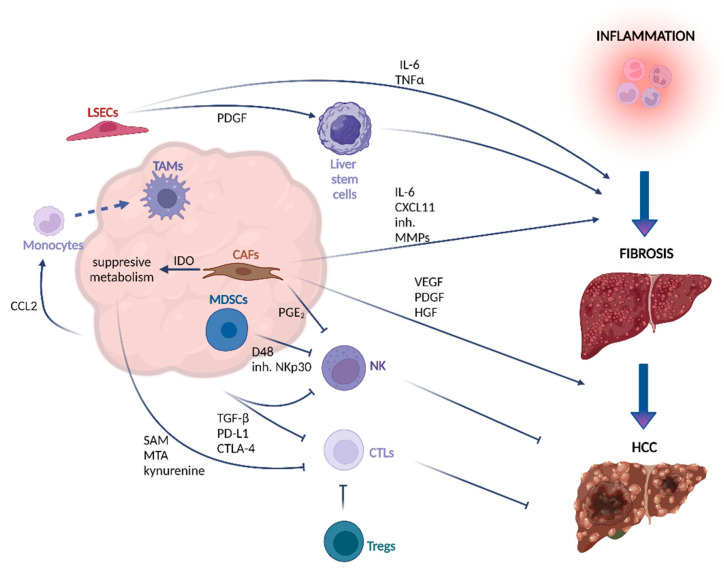
Crosstalk between the TME and HCC cells in the process of liver fibrosis and HCC development. The dynamic crosstalk between the non-cancer cells and cancer cells is crucial for the process of liver fibrosis and HCC development. Complex interactions influence cancer progression and result in inhibition of the anti-cancer response by activating and mobilizing immune cells with immunosuppressive properties. The inflammatory environment of HCC leads to upregulation of CCL2, which recruits monocytes to the tissue where they can differentiate into macrophages. LSECs stimulated by pathogen infections release the proinflammatory IL-6 and TNF-α, and initiate stem cell activation through PDGF release. The stemness of HCC cells is also supported by HGF secreted by CAFs. CAFs also promote tumorigenesis by expressing pro-angiogenic cytokines, such as VEGF and PDGF. Increased expression of CXCL11 by CAFs facilitates the recruitment of T cells into inflammatory sites, supporting the self-renewal of tumor-initiating cells. NK cell activation and cytotoxic activity is inhibited by PGE_2_ and IDO enzymes produced by CAFs, and also by blocking of the NKp30 receptor by MDSCs. TAMs express the D48 ligand, which interacts with the 2B4 receptor on NK cells, causing their exhaustion and apoptosis. High levels of TGF-β enhance the expression of inhibitory receptors, including PD-1 and CTLA-4, on T cells, impairing their proliferation, differentiation, and activation. Additionally, the release of kynurenine, SAM, and MTA by cancer cells in the TME can lead to T cell exhaustion. Tregs promote immune tolerance against neoplastic cells by suppressing CTLs. Figure created with BioRender.com (accessed on 27 February 2022).

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
