# Peer review of "Tumor Microenvironment of Hepatocellular Carcinoma: Challenges and Opportunities for New Treatment Options"

_ijms, 2022, doi:10.3390/ijms23073778_

Round 1
Reviewer 1 Report
Title: “Tumor Microenvironment of hepatocellular carcinoma:challenges and opportunities for new treatment options” Authors: Zuzanna Sas, Ewa Cendrowicz, Isabel Weinhäuser, Tomasz P. Rygiel
Summary:
In this review by Zuzanna Sas et al, the authors describe the components of the tumor microenvironment in hepatocellular carcinoma, including the specific role of tumor-associated macrophages, and provide an overview of the role of the tumor microenvironment in hepatocellular carcinoma progression and explain recent advances in therapies targeting hepatocellular carcinoma.
Several Points:
1: It should be mentioned in the abstract and at the end of the introduction whether the review refers to TME in vitro or in vivo.
2: In Chapter 3.1, reference should be made to Figure 1.
3: Following Chapter 3, a table summarizing the major components of TME and their effects would be helpful.
4: At the end of Chapter 3, there should be a summary sentence on the TME components.
5: Chapter 4 should discuss in more detail the development of resistance to standard therapies, as this is a major challenge (in line with the title). It could be mentioned here that not only are combinations of conventional drugs being tested to overcome cancer cell resistance, but also the complementary use of phytopharmaceuticals is increasingly being investigated. Sample references:
doi: 10.3390/biomedicines8080236.; doi: 10.2174/138920112798868791.
doi: 10.3390/molecules25184292.; doi: 10.1111/jfbc.13761.;
doi: 10.3389/fphar.2021.699842.
6: In the discussion, the term "combinatorial therapies" should be clarified, e.g., combining standard drugs or drugs with phytopharmaceuticals.
7: A list of key abbreviations might be helpful.
8: References should be reviewed and formatted consistently. Some are marked as incorrect in the text and on page 7 ("In a study published by Wang et al." ...) the source citation is missing. In addition, in some places several papers are cited but only one first author is named. This should be corrected, e.g., pp. 3 refs. 32, 33; p. 6 refs. 69-71.
9: Minor formatting changes such as letter errors in the abbreviation MoMos and "PD-L1/CTLA-4 inhibitor" in the first sentence in section 4.2.2.
Author Response
Reviewer 1:
1: It should be mentioned in the abstract and at the end of the introduction whether the review refers to TME in vitro or in vivo.
Thank you for comment. The real Tumor microenvironment (TME) is only in vivo (in patients). We have added information in the abstract and in the introduction referring to patients and primary HCC. In general, we tried to focus our review on the TME in vivo, however some studies cited in this review are also in vitro studies.
2: In Chapter 3.1, reference should be made to Figure 1.
We have added the reference.
3: Following Chapter 3, a table summarizing the major components of TME and their effects would be helpful.
Thank you for this suggestion. We have added a table with summary of the TME components (Table 1).
4: At the end of Chapter 3, there should be a summary sentence on the TME components.
We have added a short part summarizing the whole chapter 3.
5: Chapter 4 should discuss in more detail the development of resistance to standard therapies, as this is a major challenge (in line with the title). It could be mentioned here that not only are combinations of conventional drugs being tested to overcome cancer cell resistance, but also the complementary use of phytopharmaceuticals is increasingly being investigated. Sample references: doi: 10.3390/biomedicines8080236.; doi: 10.2174/138920112798868791. doi: 10.3390/molecules25184292.; doi: 10.1111/jfbc.13761.;doi: 10.3389/fphar.2021.699842.
We have added a chapter about development of resistance and possible combinatorial therapies with synthetic, biologic, and natural compounds that can overcome the resistance.
6: In the discussion, the term "combinatorial therapies" should be clarified, e.g., combining standard drugs or drugs with phytopharmaceuticals.
We have added a few sentences to discussion section about possible combinatorial approaches for HCC.
7: A list of key abbreviations might be helpful.
We have added a list of abbreviations below the main text
8: References should be reviewed and formatted consistently. Some are marked as incorrect in the text and on page 7 ("In a study published by Wang et al." ...) the source citation is missing. In addition, in some places several papers are cited but only one first author is named. This should be corrected, e.g., pp. 3 refs. 32, 33; p. 6 refs. 69-71.
Thank you for this comment. We have improved the formatting and added references consistently.
9: Minor formatting changes such as letter errors in the abbreviation MoMos and "PD-L1/CTLA-4 inhibitor" in the first sentence in section 4.2.2.
We have corrected abbreviations: monocyte-derived macrophages (MoMΦs), PD-1/PD-L1 interaction.
Reviewer 2 Report
This review is the therapy strategies for the tumor microenvironment, especially liver cancer. This is comprehensive and valuable for researchers on cancer or liver biology. However, some descriptions or introductions are lacking, so the authors should add sections or sentences for readers’ better understanding. For example, the in vitro models mimicking the liver tumor microenvironment have been recently noted for drug discovery. This tool must affect the fields of new treatment options. Taken together, many major revisions should be made. This paper would be re-considered only when all the comments are responded.
1.
Recently, three-dimensional culture based on the TME for new drug discovery has been noted. The title includes therapy, so the field should be introduced briefly. To reduce the authors’ burden, I suggest recent papers be added.
Review papers (for concept)
Cancers 2020, 12(10), 2754
Research papers
Lab Chip, 2018, 18, 2036-2046
Tissue Eng. Part A, 26, 2020, 1272-1282. https://doi.org/10.1089/ten.tea.2020.0095
Lab Chip, 2018, 18, 3606-3616
2.
How about the influence of TAM polarization on fibrosis or NASH?
3.
The authors should add the table for the role of cytokines, chemokines, and surface markers.
4.
References should be added correctly.
The sentence “Error! References source not found” is often indicated.
The reviewer cannot perform the review correctly.
Author Response
- Recently, three-dimensional culture based on the TME for new drug discovery has been noted. The title includes therapy, so the field should be introduced briefly. To reduce the authors’ burden, I suggest recent papers be added.
Review papers (for concept)
Cancers 2020, 12(10), 2754
Research papers
Lab Chip, 2018, 18, 2036-2046
Tissue Eng. Part A, 26, 2020, 1272-1282. https://doi.org/10.1089/ten.tea.2020.0095
Lab Chip, 2018, 18, 3606-3616
We have included short information (chapter 4) about the application of 3D culture system for drug discovery. However, we cannot elaborate further on this topic as 3D cultures and pre-clinical drug discovery is abroad topic and not the main topic of the review. The focus is placed on the treatment and clinical trials. Furthermore, to our knowledge, no 3D cell cultures were described for HCC in particular. The original articles that reviewer has referred to, are either describing 3D cultures of normal liver or 3D cultures of other than HCC cancer types.
2.How about the influence of TAM polarization on fibrosis or NASH?
We have added one paragraph describing the topic
3.The authors should add the table for the role of cytokines, chemokines, and surface markers.
We have added a table below paragraph 3
4. References should be added correctly.
The sentence “Error! References source not found” is often indicated.
The reviewer cannot perform the review correctly.
Thank you for this comment. We have adjusted the references and improved the formatting.
Round 2
Reviewer 2 Report
I recommend the publication.